# Speeding up Deep Learning Training by Sharing Weights and Then Unsharing

## Abstract

It has been widely observed that increasing deep learning model sizes often leads to significant performance improvements on a variety of natural language processing and computer vision tasks. In the meantime, however, computational costs and training time would dramatically increase when models get larger. In this paper, we propose a simple approach to speed up training for a particular kind of deep networks which contain repeated structures, such as the transformer module. In our method, we first train such a deep network with the weights shared across all the repeated layers till some point. We then stop weight sharing and continue training until convergence. The untying point is automatically determined by monitoring gradient statistics. Our adaptive untying criterion is obtained from a theoretic analysis over deep linear networks. Empirical results show that our method is able to reduce the training time of BERT by 50%.

## 1 Introduction

It has been widely observed that increasing model size often leads to significantly better performance on various real tasks, especially natural language processing and computer vision applications (Amodei et al., 2016; He et al., 2016a; Wu et al., 2016; Vaswani et al., 2017; Devlin et al., 2019; Brock et al., 2019; Raffel et al., 2020; Brown et al., 2020). However, as models getting larger, the training can become extremely resource intensive and time consuming. As a consequence, there has been a growing interest in developing systems and algorithms for efficient distributed large-batch training (Goyal et al., 2017; Shazeer et al., 2018; Lepikhin et al., 2020; You et al., 2020).

In this paper, we seek for speeding up deep learning training by exploiting unique network architectures rather than by distributed training. In particular, we are interested in speeding up the training of a special kind of deep networks which are constructed by repeatedly stacking the same layer, for example, the transformer module (Vaswani et al., 2017). We propose a simple method for efficiently training such kind of networks. In our approach, we first force the weights to be shared across all the repeated layers and train the network, and then, at some point, we stop weight sharing and continue training until convergence. The point for stopping weight sharing can be either predefined or automatically chosen by monitoring gradient statistics during training. Empirical studies show that our method can reduce the training time of BERT (Devlin et al., 2019) by 50%.

Our method is motivated by the successes of weight sharing models, in particular, ALBERT (Lan et al., 2020). It is a variant of BERT in which the weights across all the transformer layers are shared. As long as its architecture is sufficiently large, ALBERT can be comparable with or even outperform the original BERT on various downstream natural language processing benchmarks. However, when its architecture being the same as the original BERT, ALBERT performs significantly worse. Since the weights in the original BERT are not shared at all, it is natural to expect that ALBERT's performance will be improved if we stop its weight sharing at some point of training.

To make this idea work, however, we need to know when to untie the shared weights. A randomly chosen untying point will not work. We can see this from the two extreme cases: ALBERT which shares weights all the time, and BERT which has no weight sharing at all. To find an effective solution for automatic weight untying, we turn to theoretic analysis over deep linear networks (Hardt & Ma, 2017; Laurent & Brecht, 2018; Wu et al., 2019). A deep linear model is constructed by stacking a series of matrix multiplications. In its forward pass, a deep linear model is trivially equivalent to a single matrix. However, when being trained with backpropagation, its behavior is

analogous to the deep models with non-linearity but much easier to understand. Our theoretical analysis shows that, when learning a positive definite matrix (which admits an optimal solution with all layers having the same weights), training with weight sharing can bring significantly faster convergence. More importantly, our theoretical analysis leads to the adaptive weight untying rule that we need to construct our algorithm (see Algorithm 2). Empirical studies on real tasks show that our adaptive untying method can be at least as effective as using the best untying point which is obtained by running multiple experiments of which each has a different point to untie weights and then choosing the best result.

The rest of this paper is organized as follows. We present our weight sharing algorithm in Section 2. It actually contains three versions, depending on how to stop weight sharing during training. In Section 3, we present our theoretical results for positive definite deep linear models. All the proofs are deferred to the Appendix. In Section 4, we discuss related work. In Section 5, we show detailed experimental setups and results. We also provide various ablation studies on different choices in implementing our algorithm. Finally, we conclude this paper with discussions in Section 6.

## 2    ALGORITHM: SHARING WEIGHTS AND THEN UNSHARING

Assume we have a deep network which is obtained by repeatedly stacking the same neural module $n$ times, such as the transformer module in transformer models (Vaswani et al., 2017). Denote by $w_1, \ldots, w_n$ the weights of these $n$ layers. In our method, we first train the deep network with all the weights tied. Then, after a certain number of training steps, we untie the weights and further train the network until convergence. In what follows, we first present a simple version of our algorithm in which the weight untying point is predefined. Then, we move to its adaptive version in which the layers are automatically gradually untied according to gradient statistics. Finally, we discuss a simplified variant of this adaptive method which unties the layers all the once.

**Untying weights at a fixed point**. This is the simplest version of our method (Algorithm 1). We first train the deep network with all the weights tied for a fixed number of steps, and then untie the weights and continue training until convergence.

---

**Algorithm 1** SHARING WEIGHTS AND THEN UNTYING AT AT A FIXED POINT

1: **Input:** total number of training steps $T$, untying point $\tau$, learning rates $\{\alpha^{(t)}, t = 1, \ldots, T\}$
2: Randomly and equally initialize weights $w_1^{(0)}, \ldots, w_n^{(0)}$
3: **for** $t = 1$ **to** $T$ **do**
4:    **if** $t < \tau$ **then**
5:       $w_i^{(t)} = w_i^{(t-1)} - \alpha^{(t)} \times \text{mean}\left\{\text{grad}\left(\text{loss}, w_k^{(t-1)}\right), k = 1, \ldots, n\right\}, i = 1, \ldots, n$
6:    **else**
7:       $w_i^{(t)} = w_i^{(t-1)} - \alpha^{(t)} \times \text{grad}\left(\text{loss}, w_i^{(t-1)}\right), i = 1, \ldots, n$

---

Note that, from line 1 to 5, we initialize all the weights equally, and then update them using the mean of their gradients. It is easy to see that such an update is equivalent to weight sharing or tying. For the sake of simplicity, in lines 5 and 7, we only show how to update the weights using the plain (stochastic) gradient descent rule. One can replace this plain update rule with any of their favorite optimization methods, for example, the Adam optimization algorithm (Kingma & Ba, 2015).

While the repeated layers being the most natural units for weight sharing, that is not the only choice. We may view several layers together as the weight sharing unit, and share the weights across those units. The layers within the same unit can have different weights. For example, for a 24-layer transformer model, we may combine every four layers as a weight sharing unit. Thus, there will be six such units for weight sharing. Such flexibility of choosing weight sharing units allows for a balance between "full weight sharing" and "no weight sharing" at all.

**Adaptive weight untying**. The theoretical analysis in Section 3 motives us to adaptively and gradually untie weights based on the gradient correlation of adjacent layers (Algorithm 2). To implement this idea, all layers are put in the same group during initialization. Then, at any time step of the training, suppose we have groups $\mathcal{G} = \{G_1, G_2, ..., G_k\}$. For each group $G_i$, we compute the correlation

between the gradients of any two adjacent layers by

$$\text{correlation}_j^{(i)} = \frac{\left\langle g_j^{(i)}, g_{j+1}^{(i)} \right\rangle}{\|g_j^{(i)}\|\|g_{j+1}^{(i)}\|}, \quad g_j^{(i)}, g_{j+1}^{(i)} \text{ denote the gradients of two adjacent layers in } G_i. \quad (1)$$

Once $\text{correlation}_j^{(i)}$ falls below $\rho$ (0.5 as default) consecutively for a certain number of times (3 as default), we split the group $G_i$ into two subgroups, by splitting at $j$-th layer in $G_i$. The layers in the same group are updated using the gradient mean of this group as in Algorithm 1.

---

**Algorithm 2** ADAPTIVE WEIGHT UNTYING

1: **Input:** total number of training steps $T$, threshold $\rho$, learning rates $\{\alpha^{(t)}, t = 1, \ldots, T\}$
2: All layers are put in the same group, and randomly initialized with the same weights
3: **for** $t = 1$ **to** $T$ **do**
4:     **for** adjacent layers $i, i + 1$ in the same group **do**
5:         Calculate the gradient correlation as Equation 1    //Motivated by Section 3
6:         If the correlation falls below $\rho$, break the group between layer $i$ and $i + 1$
7:     Update the weights in each group using the average of gradients within each group

---

**Simplified adaptive untying**. To simplify the multi-step adaptive untying, instead of gradually splitting the groups, we may untie the weights of all layers at once. For every certain number of iteration steps, we monitor the gradient correlations as described in multi-step adaptive untying. If more than half of these layer correlations are less than the predefined threshold $\rho$ (0.5 as default) consecutively for a certain number of times (3 as default), we stop sharing weights for all layers.

## 3 THEORETIC MOTIVATION

In this section, we show how our method including the adaptive untying criterion can be motivated by analyzing the dynamics of training a deep linear network by gradient descent. At the first glance, deep learning models may look trivial since a deep linear model is just equivalent to a single matrix. However, when being trained with backpropagation, its behavior is analogous to generic deep models. Thus, analyzing deep linear models has attracted increasing interest in the theoretic research community (Hardt & Ma, 2017; Laurent & Brecht, 2018; Wu et al., 2019).

A deep linear network is a series of matrix multiplication

$$f(\mathbf{x}; W_1, \ldots, W_L) = W_L W_{L-1} \ldots W_1 \mathbf{x}, \qquad W_l \in \mathbb{R}^{d \times d}, \quad \ell = 1, \ldots, L.$$

The task is to train the deep linear network to learn a target matrix $\Phi \in \mathbb{R}^{d \times d}$. To focus on the training dynamics, we adopt the simplified objective function

$$\mathcal{R}(W_1, \ldots W_L) = \frac{1}{2} \|W_L W_{L-1} \ldots W_2 W_1 - \Phi\|_F^2.$$

Denote $\nabla_l \mathcal{R}$ to be the gradient of $\mathcal{R}$ with respect to $W_l$. We have

$$\nabla_l \mathcal{R} = \frac{\partial \mathcal{R}}{\partial W_l} = W_{L:l+1}^T (W_{L:1} - \Phi) W_{l-1:1}^T,$$

where $W_{l_2:l_1} = W_{l_2} W_{l_2-1} \ldots W_{l_1+1} W_{l_1}$. The standard gradient update is given by

$$W_l(t + 1) = W_l(t) - \eta \nabla_l \mathcal{R}(t), \quad l = 1, \ldots, L.$$

To train with weights shared, all the layers need to have the same initialization. And the update is

$$W_l(t + 1) = W_l(t) - \frac{\eta}{L} \sum_{i=1}^{L} \nabla_i \mathcal{R}(t), \quad l = 1, \ldots, L. \quad (2)$$

Since the initialization and updates are the same for all layers, the parameters $W_1(t), \ldots, W_L(t)$ are equal for all $t$. For simplicity, we denote the weight at time $t$ to be $W_0(t)$. Notice that the gradients are averaged, the norm of update to each layer doesn't scale with $L$.

Suppose that the target matrix $\Phi$ is a positive definite matrix. It is immediate that $\Phi^{1/n}$ is a solution to the deep linear network. Before looking into the detailed convergence analysis, it worth first showing a Lemma that reveals the updates in the weight sharing training.

**Lemma 1.** *With a positive definite target matrix $\Phi$ and initializing with $W_0(0) = I$, update the parameters according to Equation 2, we have*

$$W_l(t+1) - W_l(t) = -\eta W_0^{L-1}(t)\left(W_0^L(t) - \Phi\right), \quad l = 1, ..., L, \quad \forall t \geq 0.$$

Intuitively, the Lemma 1 shows that weight sharing allows all the layers to be trained "equally well", the layers that are far away from the output layer won't suffer from gradient vanishing or exploding.

In the following subsections, we first study the convergence result with continuous-time gradient descent, which demonstrates the benefit of training with weight sharing when learning a positive definite matrix $\Phi$. We then extend the results to the discrete-time gradient descent. We draw a comparison with training with zero-asymmetric (ZAS) initialization (Wu et al., 2019). To the best of our knowledge, ZAS gives the state-of-the-art convergence rate. It is actually the only work showing the global convergence of deep linear network trained by gradient descent for an arbitrary target matrix.

### 3.1 CONTINUOUS-TIME GRADIENT DESCENT

With continuous-time gradient descent (i.e. $\eta \to 0$), training with gradient descent and ZAS, the loss decays as $\mathcal{R}(t) \leq \exp(-2t)\mathcal{R}(0)$. For training with weight sharing, the loss becomes $\mathcal{R}(t) \leq \exp(-2L\min(1, \lambda_{\min}(\Phi))t)\mathcal{R}(0)$, when the target matrix $\Phi$ is positive definite. The extra $L$ in the exponent demonstrates the acceleration of training with weight sharing.

With $\eta \to 0$, the training dynamics of continuous-time gradient descent can be described as

$$\frac{dW_l(t)}{dt} = \dot{W}_l(t) = -\nabla_l \mathcal{R}(t), \quad l = 1, ..., L, \quad t \geq 0.$$

The ZAS initializes the weights $W_1 = W_2 = \cdots = W_{L-1} = I$ and $W_L = 0$. It helps avoiding saddle points and has the following convergence result.

**Theorem 1.** *[Continuous-time gradient descent without weight sharing (Wu et al., 2019)] For the deep linear network $f(\mathbf{x}; W_1, ..., W_L) = W_L W_{L-1}...W_1 \mathbf{x}$, the continuous time gradient descent with the zero-asymmetric initialization satisfies $\mathcal{R}(t) \leq \exp(-2t)\mathcal{R}(0)$.*

Theorem 1 shows that with the zero-asymmetric initialization, the continuous gradient descent linearly converges to the global optimal solution for general target matrix $\Phi$.

Next, considering the special case where the goal is to learn a positive definite matrix $\Phi$. Based on Lemma 1, we have the following convergence result for training with weight sharing.

**Theorem 2.** *[Continuous-time gradient descent with weight sharing] For the deep linear network $f(\mathbf{x}; W_1, ..., W_L) = W_L W_{L-1}...W_1 \mathbf{x}$, initialize all $W_l(0)$ with identity matrix $I$ and update according to Equation 2. With a positive definite target matrix $\Phi$, the continuous-time gradient descent satisfies $\mathcal{R}(t) \leq \exp(-2L\min(1, \lambda_{min}^2(\Phi))t)\mathcal{R}(0)$.*

**Remark 1.** *The difference between convergence rates in Theorem 1 and Theorem 2 is not an artifact of analysis. For example, when the target matrix is simply $\Phi = \alpha I, \alpha > 1$. It can be explicitly shown that with the initialization in Theorem 1, we have $\dot{\mathcal{R}}(0) = -2\mathcal{R}(0)$ while training with weight sharing (Theorem 2), we have $\dot{\mathcal{R}}(0) = -2L\mathcal{R}(0)$. This implies that the convergence results in Theorem 1 and Theorem 2 cannot be improved in general.*

The extra $L$ in the exponent leads to faster convergence. The key to show the acceleration is

$$\frac{d\mathcal{R}(t)}{dt} = \sum_{l=1}^{L} \text{tr}\left(\nabla_l^\top \mathcal{R}(t)\dot{W}_l(t)\right) \leq -2L\lambda_{\min}^2(W_0(t)^{L-1})\mathcal{R}(t),$$

where we see the $L$ comes from the summation. This sheds light on two important factors:

1. All layers need to have sufficiently large update (i.e. $\dot{W}_l(t)$ is large for all $l$).

2. Each layer's update needs to well correlate with its gradient (i.e. $\nabla_l \mathcal{R}(t)$ correlates with $\dot{W}_l(t)$).

Initializing the weights to be the same and using the average of gradients guarantees that all layers are sufficiently trained. The high correlation of $\nabla_l \mathcal{R}(t)$ and $\dot{W}_l(t)$ relies on $\Phi$ being positive definite.

Suppose the gradients of different layers do not correlate well (e.g. $\mathrm{tr}\left(\nabla_i \mathcal{R}(t) \nabla_j \mathcal{R}(t)\right) \approx 0, i \neq j$) and the weights are still forced to be shared via the updates according to Equation 2. Recall that $\dot{W}_l(t) = \frac{1}{L} \sum_{i=1}^{L} \nabla_i \mathcal{R}(t)$, we then have $\sum_{l=1}^{L} \mathrm{tr}\left(\nabla_l^\top \mathcal{R}(t) \dot{W}_l(t)\right) \approx -\frac{1}{L} \sum_{l=1}^{L} \|\nabla \mathcal{R}(t)\|_F^2$, which loses the extra $L$ acceleration in the convergence due to the $1/L$ leading factor.

When dealing with real deep learning models, there is no guarantee that all the gradients at different layers highly correlate. Thus, we may monitor gradient correlations during training: sharing weights when gradients well correlate, and break the ties when gradient correlations fall below a certain threshold. This matches the adaptive untying rule we proposed in Section 2.

## 3.2 DISCRETE-TIME GRADIENT DESCENT

Here we extend the previous result to the discrete-time gradient descent with a positive constant step size $\eta$. It can be shown that with zero-asymmetric initialization, training with the gradient descent will achieve $\mathcal{R}(t) \leq \epsilon$ within $O(L^3 \log(1/\epsilon))$ steps; initializing and training with weights sharing, the deep linear network will learn a positive definite matrix $\Phi$ to $\mathcal{R}(t) \leq \epsilon$ within $O(L \log(1/\epsilon))$ steps, which reduces the required iterations by a factor of $L^2$.

To make easy comparisons, we first repeat without proving the discrete-time gradient descent convergence result of ZAS.

**Theorem 3.** *[Continuous-time gradient descent without weight sharing (Wu et al., 2019)] For deep linear network $f(\mathbf{x}; W_1, ..., W_L) = W_L W_{L-1} ... W_1 \mathbf{x}$ with zero-asymmetric initialization and discrete-time gradient descent, if the learning rate satisfies $\eta \leq \min\left\{\left(4L^3\xi^6\right)^{-1}, \left(144L^2\xi^4\right)^{-1}\right\}$, where $\xi = \max\left\{2\|\Phi\|_F, 3L^{-1/2}, 1\right\}$, then we have linear convergence $\mathcal{R}(t) \leq \left(1 - \frac{\eta}{2}\right)^t \mathcal{R}(0)$.*

Since the learning rate is $\eta = O(L^{-3})$, Theorem 3 indicates that the gradient descent can achieve $\mathcal{R}(t) \leq \epsilon$ within $O(L^3 \log(1/\epsilon))$ steps.

In the special case of learning a positive definite matrix $\Phi$, initialize all weights $W_l$ to be the same and train with weights sharing, we have the following convergence result.

**Theorem 4.** *[Discrete-time gradient descent with weight sharing] For the deep linear network $f(\mathbf{x}; W_1, ..., W_L) = W_L W_{L-1} ... W_1 \mathbf{x}$, initialize all $W_l(0)$ with identity matrix $I$ and update according to Equation 2. With a positive definite target matrix $\Phi$, and setting $\eta \leq \frac{\min(\lambda_{min}^2(\Phi), 1)}{4\sqrt{d}L^2 \max(\lambda_{max}^4(\Phi), 1)}$, we have linear convergence $\mathcal{R}(t) \leq \exp\left[-(2L - 2)\min\left(\lambda_{min}^2(\Phi), 1\right)\eta t\right] \mathcal{R}(0)$.*

Take $\lambda_{\min}(\Phi)/\lambda_{\max}(\Phi), d$ as constants and focus on the scaling with $L, \epsilon$, we have $\eta = O(L^{-2})$. Because of the extra $L$ in the exponent, we know that when learning a positive definite matrix $\Phi$, training with weight sharing can achieve $\mathcal{R}(t) \leq \epsilon$ within $O\left(L \log\left(1/\epsilon\right)\right)$ steps. The dependency on $L$ reduces from previous $L^3$ to linear, which shows the acceleration of training by weight sharing.

## 4 RELATED WORK

Lan et al. (2020) propose ALBERT with the weights being shared across all its transformer layers. Large ALBERT models can achieve good performance on several natural language understanding benchmarks. Bai et al. (2019b) propose trellis networks which are temporal convolution networks with shared weights and obtain good results for language modeling. This line of work is then extended to deep equilibrium models (Bai et al., 2019a) which are equivalent to infinite-depth weight-tied feedforward networks. Dabre & Fujita (2019) show that the translation quality of a model that recurrently stacks a single layer is comparable to having the same number of separate layers. Zhang et al. (2020) also demonstrate the application of weight-sharing in neural architecture search.

Deep linear models have been widely studied for its simplicity and similarity to deep learning models. Baldi & Hornik (1989) show that all local minima are also global minima for two-layer linear networks. Laurent & Brecht (2018) extend the same result to deep linear networks. Hardt & Ma

(2017) show the PL condition is satisfied within the neighbour of a global optimum. Shamir (2019) show that, for one-dimensional deep linear networks, with the Xavier or near-identity initialization, it requires at least $\exp(\Omega(L))$ steps to converge, where $L$ is the depth. Wu et al. (2019) show that this result can be improved to $O(L^3 \log 1/\epsilon)$ with a special zero-asymmetric initialization.

## 5 EXPERIMENTS

In this section, we present the experimental setup and results for training the BERT Large model with the standard training procedure as in the literature as well as our Sharing WEights (SWE) method. In what follows, without explicit clarification, BERT always means the BERT Large model.

### 5.1 EXPERIMENTAL SETUP

We use the TensorFlow official implementation of BERT (team & contributors). We first show experimental results with English Wikipedia and BookCorpus for pre-training as in the original BERT paper (Devlin et al., 2019). We then move to the XLNet enlarged pretraining dataset (Yang et al., 2019). We preprocess all datasets with WordPiece tokenization (Schuster & Nakajima, 2012). We mask 15% tokens in each sequence. For experiments on English Wikipedia and BookCorpus, we randomly choose tokens to mask. For experiments on the XLNet dataset, we do whole word masking – in case that a word is broken into multiple tokens, either all tokens are masked or not masked. For all experiments, we set both the batch size and sequence length to 512.

We use the AdamW optimizer (Loshchilov & Hutter, 2019) with the weight decay rate being 0.01, $\beta_1 = 0.9$, and $\beta_2 = 0.999$. For English Wikipedia and BookCorpus, we use Pre-LN (He et al., 2016b; Wang et al., 2019b) instead of the original BERT's Post-LN. Note that the correct implementation of Pre-LN contains a final layer-norm right before the final classification/masked language modeling layer. Unlike the claim made by Xiong et al. (2020), we notice that using Pre-LN with learning rate warmup leads to better performance. In our implementation, the learning rate starts from 0.0, linearly increases to the peak value $3 \times 10^{-4}$, the learning rate used by Xiong et al. (2020), at the $10k$-th iteration, and then linearly decays to 0.0. For the XLNet dataset, we apply the same Pre-LN setup except the peak learning chosen to be $2 \times 10^{-4}$. The peak learning rate of $3 \times 10^{-4}$ makes training unstable here and yields worse performance than $2 \times 10^{-4}$.

After pre-training, we fine-tune the models for the Stanford Question Answering Dataset (SQuAD v1.1 and SQuAD v2.0) (Rajpurkar et al., 2016) and the GLUE benchmark (Wang et al., 2019a). For all fine-tuning tasks, we follow the setting as in the literature: the model is fine-tuned for 3 epochs; the learning rate warms up linearly from 0.0 to peak in the first 10% of the training iterations, then linearly decay to 0.0. We select the best peak learning rate based on the validation set from $\{1 \times 10^{-5}, 1.5 \times 10^{-5}, 2 \times 10^{-5}, 3 \times 10^{-5}, 4 \times 10^{-5}, 5 \times 10^{-5}, 7.5 \times 10^{-5}, 10 \times 10^{-5}, 12 \times 10^{-5}\}$. For the SQuAD datasets, we fine-tune each model 5 times and report the average. For the GLUE benchmark, for each training method, we simply train one BERT model and submit the model's predictions over the test sets to the GLUE benchmark website to obtain test results. We observed that when using Pre-LN, the GLUE finetuning process is stable and no model diverged.

**Training methods.** The training procedure in the TensorFlow official implementation of BERT serves as our baseline, where the baseline training takes 1 million steps (both on English Wikipedia plus BookCorpus and on the enlarged XLNet dataset). For our Sharing WEight (SWE) method, only half of the number of iterations is taken. For a complete comparison, we also report the results from the baseline method with half of the number of iterations. Three versions of our method with hyperparameter settings are listed below.

**SWE-F** **F**ixed point untying. It serves as a baseline of its adaptive version. We ran experiments with a set of different untying points, and identified the best untying point $\tau = 50k$. The full results from different $\tau$ values are presented in Section 5.3.1.

**SWE-A** **A**daptive untying. We check the gradient correlations for every 1000 iterations. If the gradient correlation of adjacent layers is below a threshold $\rho = 0.5$ for three consecutive times, we break the tie. The effect of different $\rho$ values is studied in Section 5.3.1.

**SWE-S** **S**implified adaptive untying. We use the same setup as in SWE-A, except we break the tie all at once, when the majority of gradient correlations are below the threshold $\rho$ for three consecutive times.

## 5.2 EXPERIMENT RESULTS

For English Wikipedia and BookCorpus, both pretraining and finetuning results of our method vs. the baseline method are shown in Table 1. From the results, we see that our method with 500k training iterations matches the performance of the baseline method with 1 million training iterations, and significantly outperforms the baseline method with 500k training iterations. The results for the XLNet dataset are shown in Table 2. We observe similar advantages of our approach over the baseline.

Table 1: Training BERT on English Wikipedia and BookCorpus. Our method with half of a million iterations matches the baseline performance with one million iteration steps, and outperforms the baseline with half of a million iterations.

|  | Baselines | | Our method, **0.5m iter.** | | |
| --- | --- | --- | --- | --- | --- |
|  | **1m iter.** | **0.5m iter.** | SWE-F | SWE-A | SWE-S |
| Pretrain MLM (acc.%) | 74.98 | 73.66 | 73.92 | 74.03 | 74.13 |
| SQuAD v1.1 (F-1%) | 92.58 | 91.54 | 92.54 | 92.31 | 92.54 |
| SQuAD v2.0 (F-1%) | 85.06 | 82.99 | 84.14 | 84.16 | 84.79 |
| GLUE/AX (corr%) | 42.3 | 39.0 | 40.1 | 42.4 | 41.8 |
| GLUE/MNLI-m (acc.%) | 86.9 | 85.9 | 87.2 | 87.4 | 87.2 |
| GLUE/MNLI-mm (acc.%) | 86.1 | 85.4 | 86.7 | 86.8 | 86.7 |
| GLUE/QNLI (acc.%) | 93.5 | 92.3 | 93.4 | 93.2 | 92.9 |
| GLUE/QQP (F-1%) | 72.2 | 72.1 | 71.7 | 71.8 | 71.5 |
| GLUE/SST-2 (acc.%) | 94.8 | 94.3 | 94.8 | 95.6 | 94.2 |

Table 2: Training BERT on the XLNet dataset. Our method with half of a million iterations matches the baseline performance with one million iteration steps, and outperforms the baseline performance with half of a million iterations.

|  | Baselines | | Our method, **0.5m iter.** | | |
| --- | --- | --- | --- | --- | --- |
|  | **1m iter.** | **0.5m iter.** | SWE-F | SWE-A | SWE-S |
| Pretrain MLM (acc.%) | 71.75 | 70.06 | 70.18 | 70.92 | 70.60 |
| SQuAD v1.1 (F-1%) | 93.37 | 92.41 | 92.82 | 92.65 | 92.99 |
| SQuAD v2.0 (F-1%) | 86.49 | 85.28 | 85.57 | 85.41 | 86.12 |
| GLUE/AX (corr%) | 44.2 | 43.2 | 43.7 | 44.1 | 43.3 |
| GLUE/MNLI-m (acc.%) | 88.6 | 87.3 | 88.4 | 87.7 | 88.2 |
| GLUE/MNLI-mm (acc.%) | 87.9 | 87.4 | 87.4 | 87.4 | 87.6 |
| GLUE/QNLI (acc.%) | 92.3 | 91.9 | 92.8 | 92.8 | 93.2 |
| GLUE/QQP (F-1%) | 72.3 | 71.7 | 72.1 | 72.2 | 72.0 |
| GLUE/SST-2 (acc.%) | 95.9 | 95.7 | 95.4 | 95.9 | 96.0 |

## 5.3 ABLATION STUDIES

In this section, we study the effects of different choices in implementing our method.

### 5.3.1 WHEN TO STOP WEIGHT SHARING

In this section, we study the effects of using different untying points (Algorithm 1) and thresholds (Algorithm 2). If weights are shared throughout the entire pretraining process, the final performance

will be much worse than without any form of weight sharing (Lan et al., 2020). On the other hand, without weight sharing at all yields slower convergence.

Results of using different untying point $\tau$ and threshold $\rho$ values are summarized in Table. 3. Models are trained for 500k iterations on English Wikipedia and BookCorpus. From the results, we see that for the SWE-F method, a smaller $\tau$ value performs better than a larger $\tau$. This means that the weight sharing stage should not be too long. We also see that the performance of the SWE-A method is not sensitive to the threshold $\rho$.

Table 3: Results from different untying points $\tau$ and thresholds $\rho$. Models are trained for 500k iterations on English Wikipedia and BookCorpus.

|  | SWE-F | | | SWE-A | | | | |
|---|---|---|---|---|---|---|---|---|
|  | $\tau$=25k | $\tau$=50k | $\tau$=200k | $\rho$=0.1 | $\rho$=0.3 | $\rho$=0.5 | $\rho$=0.7 | $\rho$=0.9 |
| Pretrain MLM (acc.%) | 71.48 | 73.92 | 72.42 | 73.76 | 74.11 | 74.03 | 74.12 | 73.71 |
| SQuAD v1.1 (F-1%) | 90.51 | 92.54 | 91.65 | 92.29 | 92.15 | 92.31 | 92.16 | 92.17 |
| SQuAD v2.0 (F-1%) | 81.11 | 84.14 | 83.35 | 84.69 | 84.30 | 84.16 | 83.38 | 84.21 |
| GLUE/AX (corr%) | 37.0 | 40.1 | 42.6 | 40.0 | 41.5 | 42.4 | 40.2 | 40.3 |
| GLUE/MNLI-m (acc.%) | 86.1 | 87.2 | 85.5 | 86.9 | 87.4 | 87.4 | 87.0 | 86.5 |
| GLUE/MNLI-mm (acc.%) | 83.9 | 86.7 | 85.4 | 86.1 | 86.4 | 86.8 | 86.9 | 86.6 |
| GLUE/QNLI (acc.%) | 92.8 | 93.4 | 91.9 | 93.3 | 93.3 | 93.2 | 93.3 | 92.7 |
| GLUE/QQP (F-1%) | 70.3 | 71.7 | 70.8 | 71.4 | 71.6 | 71.8 | 71.9 | 72.0 |
| GLUE/SST-2 (acc.%) | 93.4 | 94.8 | 93.5 | 93.4 | 94.7 | 95.6 | 95.2 | 94.9 |

### 5.3.2 How to choose weight sharing units

Note that it is not necessary to be restricted to share weights only across the original layers. We can group several consecutive layers as a weight sharing unit. We denote $A \times B$ as grouping $A$ layers as a weight sharing unit which is being shared with $B$ times. Since BERT has 24 layers, the baseline method without weight sharing can be viewed as "24x1", and our method shown in Table 1 can be viewed as "1x24". We present results from more different choices of weight sharing units in Table 4. We can see that, in order to achieve good results, the size of the chosen weight sharing unit should not be larger than 6 layers. This means that the weights of a layer must be shared for at least 4 times.

Table 4: We group several consecutive layers as a weight sharing unit instead of sharing weights only across original layers. $A \times B$ means grouping $A$ layers as a unit which is being shared with $B$ times. Models are trained for 500k iterations on English Wikipedia and BookCorpus.

|  | Baseline | SWE-F | | | | |
|---|---|---|---|---|---|---|
|  | 24x1 | 12x2 | 6x4 | 4x6 | 2x12 | 1x24 |
| Pretrain MLM (acc.%) | 73.66 | 73.82 | 73.99 | 73.90 | 74.16 | 73.92 |
| SQuAD v1.1 (F-1%) | 91.54 | 92.18 | 92.62 | 92.52 | 92.44 | 92.54 |
| SQuAD v2.0 (F-1%) | 82.99 | 83.74 | 84.56 | 85.82 | 85.06 | 84.14 |
| GLUE/AX (corr%) | 39.0 | 40.7 | 40.3 | 40.2 | 43.0 | 40.1 |
| GLUE/MNLI-m (acc.%) | 85.9 | 86.9 | 87.9 | 87.0 | 87.1 | 87.2 |
| GLUE/MNLI-mm (acc.%) | 85.4 | 85.8 | 87.0 | 86.4 | 86.4 | 86.7 |
| GLUE/QNLI (acc.%) | 92.3 | 93.1 | 93.2 | 92.9 | 93.8 | 93.4 |
| GLUE/QQP (F-1%) | 72.1 | 72.0 | 72.0 | 71.6 | 71.8 | 71.7 |
| GLUE/SST-2 (acc.%) | 94.3 | 94.7 | 94.3 | 94.8 | 94.7 | 94.8 |

## 6 Conclusion

We proposed a simple weight sharing method to speed up the training of deep networks with repeated layers and showed promising empirical results on BERT training. Our method is motivated by the successes of weight sharing models in the literature as well as our theoretic analysis on deep linear

models. For future work, we will extend our empirical studies to other deep learning models and tasks, and analyze under which conditions our method will be helpful.

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

# A    PROOFS OF SECTION 3

## A.1    PROOF OF LEMMA 1

*Proof.* By definition, we have

$$\nabla_l \mathcal{R}(t) = W_0(t)^{L-l} \left( W_0^L(t) - \Phi \right) W_0^{l-1}(t).$$

To see the result in Lemma 1, it is sufficient to show that $W_0(t)$ has the same eigenvectors as $\Phi$.

First, it is not hard to see that by initializing $W_0 = I$, the requirement trivially holds for $W_0(0)$. Now, suppose $W_0(t_0)$ has the same eigenvectors as $\Phi$, then from the definition of update

$$\nabla_l \mathcal{R}(t_0) = W_0^{L-l}(t_0) \left( W_0^L(t_0) - \Phi \right) W_0^{l-1}(t_0).$$

We thus know that $\nabla_l \mathcal{R}(t_0)$ has the same eigenvectors as $\Phi$ for all layer $l$. Therefore the update for $W_0(t_0)$, which is given by $\sum_{i=1}^L \nabla_i \mathcal{R}(t_0)/L$ also has the same eigenvectors as $\Phi$. Inductively, it indicates that from time step $t_0$, all subsequent weights $W_0(t)$ has the same eigenvectors as $\Phi$.

We can thus exchange the order of matrix multiplication and for all $t \geq 0$, we have

$$\nabla_l \mathcal{R}(t) = W_0(t)^{L-l} \left( W_0^L(t) - \Phi \right) W_0^{l-1}(t) = W_0^{L-1}(t) \left( W_0^L(t) - \Phi \right),$$

which directly implies that

$$W_l(t+1) - W_l(t) = -\frac{\eta}{L} \sum_{i=1}^L \nabla_i \mathcal{R}(t) = -\eta W_0^{L-1}(t) \left( W_0^L(t) - \Phi \right).$$

∎

## A.2    PROOF OF THEOREM 2

*Proof.* From Lemma 1, we have

$$\dot{W}_l(t) = -\sum_{l=1}^L \nabla_l \mathcal{R}(t)/L = -W_0^{L-1}(t) \left( W_0^L(t) - \Phi \right).$$

For the loss function $\mathcal{R}(t)$, we have

$$\dot{\mathcal{R}}(t) = \sum_{l=1}^L \mathrm{tr} \left( \nabla_l^\top \mathcal{R}(t) \dot{W}_l(t) \right) = -\sum_{l=1}^L \|\nabla_l \mathcal{R}(t)\|_F^2$$

$$= -\sum_{l=1}^L \left\| W_0(t)^{L-1} \left( W_0(t)^L - \Phi \right) \right\|_F^2$$

$$\leq -2L\lambda_{\min}^2 (W_0(t)^{L-1}) \mathcal{R}(t).$$

By continuous gradient descent with $W_0(0) = I$, it is easy to see that

$$\lambda_{\min}(W_0^{L-1}(t)) \geq \min(1, \lambda_{\min}(\Phi)).$$

Therefore we have

$$\dot{\mathcal{R}}(t) \leq -2L \min(1, \lambda_{\min}^2(\Phi)) \mathcal{R}(t) \implies \mathcal{R}(t) \leq e^{-2L \min(1, \lambda_{\min}^2(\Phi))t} \mathcal{R}(0).$$

∎

## A.3    TECHNICAL LEMMA

**Lemma 2.** *Initializing $W_0 = I$ and training with weight sharing update (Equation 2), by setting $\eta \leq \frac{1}{L \max(\lambda_{max}^2(\Phi), 1)}$, we have*

$$\lambda_{min}(W_0(t)) \geq \min(\lambda_{min}(\Phi)^{1/L}, 1), \quad \lambda_{max}(W_0(t)) \leq \max(\lambda_{max}(\Phi)^{1/L}, 1).$$

*Proof.* In the proof of Lemma 1, we know that $W_0(t)$ has the same eigenvectors as $\Phi$. Take any eigenvector, denote $\widehat{\lambda}(t)$ and $\lambda^L$ to be the corresponding eigenvalue of $W_0(t)$ and $\Phi$. By Lemma 1, we have

$$\widehat{\lambda}(t+1) = \widehat{\lambda}(t) - \eta\widehat{\lambda}(t)^{L-1}\left(\widehat{\lambda}(t)^L - \lambda^L\right).$$

For $\lambda > 1$, we would like to show $\widehat{\lambda}(t) \in [1, \lambda], \forall t \geq 0$ by setting $\eta \leq \frac{1}{L\lambda^{2L}}$. Since $\widehat{\lambda}(0) = 1$, we know this claim holds trivially at $t = 0$. Then suppose we have the claim holds for $t = t_0$, then

$$\widehat{\lambda}(t_0+1) = \widehat{\lambda}(t_0) - \eta\widehat{\lambda}(t_0)^{L-1}\left(\widehat{\lambda}(t_0) - \lambda\right)\sum_{i=0}^{L-1}\widehat{\lambda}(t_0)^i\lambda^{L-1-i}.$$

To make $\widehat{\lambda}(t_0+1) \in [1, \lambda]$, we set $\eta \leq \frac{1}{L\lambda^{2L}}$ and have

$$\widehat{\lambda}(t_0+1) \leq \widehat{\lambda}(t_0) - \frac{\widehat{\lambda}(t_0)^{L-1}\sum_{i=0}^{L-1}\widehat{\lambda}(t_0)^i\lambda^{L-1-i}}{L\lambda^{2L}}\left(\widehat{\lambda}(t_0) - \lambda\right)$$

$$\leq \widehat{\lambda}(t_0) - \frac{\widehat{\lambda}(t_0)^{L-1}\sum_{i=0}^{L-1}\widehat{\lambda}(t_0)^i\lambda^{L-1-i}}{\lambda^{L-1}\sum_{i=1}^{L-1}\lambda^{L-1}}\left(\widehat{\lambda}(t_0) - \lambda\right) \leq \lambda.$$

And $\eta \geq 0$ guarantees that $\widehat{\lambda}(t_0+1) \geq \widehat{\lambda}(t_0) \geq 1$. By induction, when $\lambda > 1$, we have $\widehat{\lambda}(t) \in [1, \lambda], \forall t \geq 0$.

Similarly, for $\lambda < 1$, we would like to show $\widehat{\lambda}(t) \in [\lambda, 1]$ by setting $\eta \leq \frac{1}{L}$. Note again the claim holds trivially when $t = 0$. Suppose $\widehat{\lambda}(t_0) \in [\lambda, 1]$, we have

$$\widehat{\lambda}(t_0+1) \geq \widehat{\lambda}(t_0) - \frac{\widehat{\lambda}(t_0)^{L-1}\sum_{i=0}^{L-1}\widehat{\lambda}(t_0)^i\lambda^{L-1-i}}{1\sum_{i=0}^{L-1}1}\left(\widehat{\lambda}(t_0) - \lambda\right) \geq \lambda.$$

And $\eta \geq 0$ guarantees that $\widehat{\lambda}(t_0+1) \leq \widehat{\lambda}(t_0) \geq 1$. By induction, when $\lambda < 1$, we have $\widehat{\lambda}(t) \in [\lambda, 1], \forall t \geq 0$.

Note that the two claims hold for all $\lambda$, it then directly implies that by setting $\eta \leq \frac{1}{L\max(\lambda_{\max}^2(\Phi),1)}$, we have

$$\lambda_{\min}\left(W_0(t)\right) \geq \min(\lambda_{\min}(\Phi)^{1/L}, 1), \quad \lambda_{\max}\left(W_0(t)\right) \leq \max(\lambda_{\max}(\Phi)^{1/L}, 1),$$

which completes the proof. ∎

### A.4 PROOF OF THEOREM 4

*Proof.* Denote $\phi = \max(\lambda_{\max}(\Phi), 1)$. From the proof of Lemma 1, we have

$$\nabla_l\mathcal{R}(t) = W_0^{L-1}(t)\left(W_0^L(t) - \Phi\right),$$

By Lemma 2, setting $\eta \leq \frac{1}{L\max(\lambda_{\max}^2(\Phi),1)}$, we have

$$\lambda_{\min}\left(W_0(t)\right) \geq \min(\lambda_{\min}(\Phi)^{1/L}, 1), \quad \lambda_{\max}\left(W_0(t)\right) \leq \max(\lambda_{\max}(\Phi)^{1/L}, 1).$$

Denote $\phi = \max(\lambda_{\max}(\Phi), 1)$, we immediately have

$$\|\nabla_l\mathcal{R}(t)\|_F \leq \phi^{(L-1)/L}\sqrt{2\mathcal{R}(t)}, \quad \|\nabla_l\mathcal{R}(t)\|_F \geq \min(\lambda_{\min}(\Phi), 1)^{(L-1)/L}\sqrt{2\mathcal{R}(t)}.$$

With one step of gradient update, we have

$$\mathcal{R}(t+1) - \mathcal{R}(t) = \frac{1}{2}\left[\|\left(W_0 - \eta\nabla_0\mathcal{R}(t)\right)^L - \Phi\|_F^2 - \|W_0 - \Phi\|_F^2\right]$$

$$= \left(\left(W_0 - \eta\nabla_0\mathcal{R}(t)\right)^L - W_0^L\right)\odot\left(W_0^L - \Phi\right) + \frac{1}{2}\|\left(W_0 - \eta\nabla_0\mathcal{R}(t)\right)^L - W_0^L\|_F^2,$$

where the $\odot$ denotes the element-wise multiplication. Let

$$I_1 = \eta A_1 \odot \left(W_0^L - \Phi\right), \quad I_2 = \sum_{k=2}^{L}\eta^k A_k \odot \left(W_0^L - \Phi\right), \quad I_3 = \frac{1}{2}\|\sum_{k=1}^{L}\eta^k A_k\|_F^2,$$

where the matrix $A_k$ comes from

$$(W_0 - \eta \nabla_0 \mathcal{R}(t))^L = A_0 + \eta A_1 + \cdots + \eta^L A_L.$$

We have

$$\mathcal{R}(t+1) - \mathcal{R}(t) \le I_1 + I_2 + I_3.$$

Note that

$$\|W_0\|_2 \le \phi^{1/L}, \quad \|\nabla_0 \mathcal{R}(t)\|_F \le \phi^{(L-1)/L} \sqrt{2\mathcal{R}(t)}.$$

Using the fact that for $0 \le y \le x/L^2$,

$$(x+y)^L \le x^L + 2Lx^{L-1}y, \quad (x+y)^L \le x^L + Lx^{L-1}y + L^2 x^{L-2} y^2.$$

Take $\eta \le \frac{1}{L^2 \phi \sqrt{2\mathcal{R}(t)}}$, we have

$$\|\sum_{k=1}^{L} \eta^k A_k\|_F \le \left( \phi^{1/L} + \eta \phi^{\frac{L-1}{L}} \sqrt{2\mathcal{R}(t)} \right)^L - \phi \le 2\eta L \phi^2 \sqrt{2\mathcal{R}(t)}$$

$$\|\sum_{k=2}^{L} \eta^k A_k\|_F \le \left( \phi^{1/L} + \eta \phi^{\frac{L-1}{L}} \sqrt{2\mathcal{R}(t)} \right)^L - \phi - \eta L \phi^{2(L-1)/L} \sqrt{2\mathcal{R}(t)} \le 2\eta^2 L^2 \phi^3 \mathcal{R}(t).$$

Thus we have

$$I_2 \le \|\sum_{k=2}^{L} \eta^k A_k\|_F \|W_0 - \Phi\|_F \le 2\sqrt{2}\eta^2 L^2 \phi^2 \mathcal{R}(t)^{3/2}$$

$$I_3 \le \frac{1}{2} \|\sum_{k=1}^{L} \eta^k A_k\|_F^2 \le 4\eta^2 L^2 \phi^4 \mathcal{R}(t).$$

For $I_1$, we directly have

$$I_1 = -\eta L \nabla_0 \mathcal{R}(t) \odot W_0^{L-1} \left( W_0^L - \Phi \right) = -\eta L \|\nabla_0 \mathcal{R}(t)\|_F^2.$$

Since we have $\|\nabla_0 \mathcal{R}(t)\|_F \ge \min \left( \lambda_{\min}(\Phi), 1 \right) \sqrt{2\mathcal{R}(t)}$. Therefore by setting

$$\eta \le \min \left( \frac{1}{L \max \left( \lambda_{\max}^2(\Phi), 1 \right)}, \frac{1}{\sqrt{2}L^2 \phi \mathcal{R}(t)^{1/2}}, \frac{\min(\lambda_{\min}^2(\Phi), 1)}{2\sqrt{2}L^2 \phi^2 \mathcal{R}(t)^{1/2}}, \frac{\min(\lambda_{\min}^2(\Phi), 1)}{4L^2 \phi^4} \right).$$

We have

$$\mathcal{R}(t+1) - \mathcal{R}(t) \le (2 - 2L)\eta \min(\lambda_{\min}^2(\Phi), 1) \mathcal{R}(t).$$

By directly setting

$$\lambda \le \frac{\min(\lambda_{\min}^2(\Phi), 1)}{4\sqrt{d}L^2 \max \left( \lambda_{\max}^4(\Phi), 1 \right)},$$

we can satisfy all the requirements above, which will give us

$$\mathcal{R}(t) \le \left[ 1 - (2L - 2) \min \left( \lambda_{\min}^2(\Phi), 1 \right) \eta \right]^t \mathcal{R}(0) \le \exp \left[ -(2L-2) \min \left( \lambda_{\min}^2(\Phi), 1 \right) \eta t \right] \mathcal{R}(t).$$

∎

