# OpenReview forum: "Speeding up Deep Learning Training by Sharing Weights and Then Unsharing"
_ICLR.cc/2021/Conference — Reject_

### Official Review · AnonReviewer1 · 2020-10-26
**Simple yet effective method, deeper analysis is needed**

**Rating:** 5
**Confidence:** 3

**Review:**

This paper proposes a simple yet effective method to train transformer-like networks for NLP tasks. Weight sharing idea is not new, it has been investigated in ALBERT as mentioned in the paper and CV areas (see refs [1],[2]).  Though novelty of this paper is not strong enough, their experimental results are adequate to show the effectiveness of proposed method.  The writing and description are quite clear, but you need to clarify the definition of 'iters' you used in  SWE exps, does it mean iters_of_weight_share + iters_of_wight_unshare?

To show the advantage of weight sharing, the authors take deep linear model as an example to show that with weight sharing, layers far away from output will not suffer from gradient vanishing or explosion, which is true for deep LINEAR models. But for deep learning based models, non-linear operations is required, which still leads to gradient vanishing or explosion even with weight sharing. For example, recurrent neural network is a typical kind of weight-sharing network and its weight is shared through time. Even with single hidden layer, RNN still suffers from gradient vanishing or explosion, and that is why LSTM is proposed in 1990s. So in my opinion, you'd better take a TRUE deep learning model as an example to analyze the effectiveness of weight sharing during training. For example, you can try to analyze deep feed forward neural networks with ReLU activation, which can be treated as a deep piece-wise linear model.

[1] J. Wang and X. Hu, “Gated recurrent convolution neural network for ocr,” in NIPS, 2017.
[2] M. Liang and X. Hu, “Recurrent convolutional neural network for object recognition,” in CVPR, 2015, pp. 3367–3375

---

> ### Author Response · Authors · 2020-11-23
> **Response to Reviewer #1**
>
> Thanks for your feedback.
>
>
> --- The “iters” in SWE experiments stands for iters_of_weight_share + iters_of_weight_unshare. The \tau in SWE-F stands for the iters_of_weight_share.
>
> --- Only weight sharing will NOT speed up training much unless we know when to untie the weights. How to automatically untie the shared weights is the whole point in this paper.  We don't claim weight sharing as our innovation here which has been invented since CNN/RNN.  What we claim here is *unsharing* in an adaptive way.
>
> --- Please see "Common Response and Revision" on our theory and its connection to the proposed algorithm. Our goal in this paper is to develop a practically useful fast training algorithm for BERT-like models. It is a simple approach which can be implemented in a few lines of code while reducing BERT training time up to 50%.  The theoretic analysis in section 3 is used to *motivate* the adaptive untying rule used in our method, which plays the most fundamental role for the success of our method.  We do *not* attempt to develop or claim any theoretical research on BERT or transformer, which is entirely different from our focus here and should be a separate research topic. On the other hand, if you think the improvement from our method is not big enough and have ideas to further improve this method, we are happy to know.

---

### Official Review · AnonReviewer4 · 2020-10-28
**Maybe a practical way to train deep networks w/ repeated layers**

**Rating:** 5
**Confidence:** 4

**Review:**

This paper provides a simple way to speed up training deep networks which contain repeated structures, such as the transformer module, by sharing the weights first and then unsharing. Three methods are shown to stop weight sharing: 1) stop weight sharing at a predefined point; 2) stop weight sharing at a point using an adaptive way, in which unsharing will be triggered when the majority of the gradient correlations between layers is below a threshold; 3) gradually untie weights between layers by splitting groups based on the gradient correlations. The authors did experiments on BERT large for SQuAD and the GLUE benchmark, and showed that their approach outperform the baseline at the same iteration of 500k, and is on par w/ the baseline training at 1M iterations.

Basically I think this approach could be a practical way to train deep networks w/ repeated layers. It's like between BERT large and ALBERT, the results are not that surprising. The main contribution of this work is that it provides the empirical results of this training method on several tasks.

My questions regarding to this paper are:

1. In Section 3, the authors provide analysis of weighting sharing can achieve better convergence rate and for good weight sharing each layer's update needs to well correlate with its gradients for deep linear network. How about any theoretical insights for networks w/ activation functions? And How could different activation functions matter (ReLU, Swish et al)? And will these activation functions affect the parameters of when to stop weight sharing?

2. For untie weights between adjacent layers, SWE-A and SWE-M, what's the motivation of setting a unified threshold for all layers? Basically I'm wondering what's the gradient correlation between adjacent layers of top layers and bottom layers, are the correlations on the same level? Will we get better accuracy if we do not limit to a unified threshold? Say, for the top 12 layers, we set the threshold to X and for the bottom 12 layers we set the threshold to Y. Need further information on this.

3. In Table 2, why is SWE-A missing?

4. For Table 3, I think the authors need to add the experiments when \tau is smaller than 50k. Basically the experiments will show that \tau should be carefully designed, an arbitrary number may not work, we still need to train some iterations w/ weight sharing for the good model performance.

5. This paper may need to cite some other papers discussing some insights of weight sharing, such as  DEEPER INSIGHTS INTO WEIGHT SHARING IN NEURAL ARCHITECTURE SEARCH (https://arxiv.org/pdf/2001.01431.pdf).

---

> ### Author Response · Authors · 2020-11-23
> **Response to Reviewer #4**
>
> Thanks for your feedback.
>
> --- On the analysis of the deep linear model
>
> Please look at "Common Response and Revision" on the top of this page. Our goal in this paper is to provide a practically useful algorithm for training BERT like models. The analysis of the deep linear models is served as a motivation for both weight sharing and using gradient correlation as the untying rule. Theoretic analysis of BERT models is not our goal here.  We didn't attempt to establish or claim such a result.
>
> --- The key contribution of our work
>
> The key contribution here  is the "adaptive weight untying"  algorithm (see algorithm 2). A random chosen untying point will NOT speed up training.
>
> --- The setting of a unified threshold
>
> We used the same threshold (=0.5) across all layers and all experiments. Experiments show that this threshold works fairly  well. Moreover, we observed that the gradients correlations all start from almost 1 and all gradually drop to 0. So, eventually all layers are untied. Layer dependent thresholds might work better but we have no idea on how to predefine them. If you have concrete ideas here, we are happy to know.
>
> --- About \tau
>
> Please note that our adaptive untying algorithm (algorithm 2)  does not need \tau. The fixed point untying method (algorithm 1) with \tau is used as a baseline of its adaptive version. Experiments show that the adaptive method can match (even outperform) the performance of the best manually chosen untying point (ran experiments with different untying points and then chose the best one to compare. See also our ablation study section). This shows our adaptive weight sharing method is effective.
>
> --- Thanks for pointing out the relevant paper. We will cite it and add discussions.
>
> --- Please see the updated version for complete experimental results.

---

### Official Review · AnonReviewer2 · 2020-10-28

**Rating:** 4
**Confidence:** 3

**Review:**

##########################################################################

Summary:

The paper proposed a new way for training models that stack the same basic block for multiple times -- share the weights first and then untie the weights. The author tried to provide theoretical insights on why it can be better. Also, ablation study shows that the proposed algorithm has marginal improvement over the baseline.

##########################################################################

Reasons for score:


The improvement over the baseline method, which just trains BERT for more steps, is quite marginal. In addition, after the weights are untied, the training process becomes exactly the same as that in BERT. It's highly likely that both methods can have similar performance after being trained for a large number of steps. Also, I do not think the theoretical results on a deep linear network can help explain the  phenomena we see in BERT training. BERT uses a much more complicated building block that involves layer normalization, attention and non-linearity. Due to these reasons, I vote for rejection.

##########################################################################

Pros:


1. The idea of first train the model with shared weights and then untie the weights is interesting.


##########################################################################

Cons:


1. The improvement over the baseline method is not very substantial. Basically, it is possible that both methods perform similarly after being trained for a longer number of steps (e.g., 1M, 1.5M).

2. I'm not convinced by the theoretical analysis. Building block in BERT is much more complicated that the building block discussed in Section 3.

3. The criteria of changing from sharing to unsharing is largely heuristic.


##########################################################################

Questions during rebuttal period:


Please address and clarify the cons above


#########################################################################

Typos:

(1) Page 3, "To be best of" should be "To the best of"

---

> ### Author Response · Authors · 2020-11-23
> **Response to Reviewer #2:  Please provide more concrete suggestions to help us improve this work**
>
> Thanks for your feedback.
>
> --- Could you please explain why  the improvement of reducing training BERT time up to 50% in our paper is considered to be "quite marginal" in your feedback. We will really appreciate if you could point out any faster training algorithm for BERT that we can compare with. We want to understand how big speed-up will be considered as "not marginal" according to your standard.
>
> --- Your argument  "it is possible that both methods perform similarly after being trained for a longer number of steps (e.g., 1M, 1.5M)." is unclear for us. It is a standard procedure in the literature (e.g., RoBERTa) to train BERT 1M steps to achieve its SODA results on NLP benchmarks. Now, based on our method,  we just need 0.5 M training steps to achieve similar results.  Don't you think this comparison is fair enough?
>
> ---  Our theoretic analysis part is conducted for motivating and developing the *adaptive untying criterion* used in our algorithm.   Our goal in this paper is to develop a practically useful algorithm to train BERT-like models faster. We do not attempt to build  or claim convergence analysis of the BERT model, which is entirely different from our focus here.  Please check the 4th paragraph in the introduction section of the  revised version for a high-level description on the connection between the theory and algorithm.  We also rewrote the algorithm section to make the connection between the theory and algorithm clearer. Moreover, the title of section 3  was changed from “theoretical analysis” to “theoretical motivation” to avoid any potential confusion or misunderstanding.

---

> > ### Comment · AnonReviewer2 · 2020-11-25
> > **Response**
> >
> > Thanks for addressing my concerns. I can understand that by just training for 0.5M steps, SWE can achieve better performance than the normal training pipeline. However, I'd like to also see whether SWE can still be better than BERT when trained for 1M steps. In fact, by looking at the MLM acc, the 0.5M + SWE still underperforms the 1.0M + normal training. In ALBERT, the author has compared the model trained for 1M and 1.5M steps and showed that the 1.5M-step model outperforms the 1M-step model. Long-time pretraining has shown to lead better models so I will need to see whether SWE can outperform BERT when trained for a longer time.
> >
> > Moreover, it is possible that the current learning rate choices: 3×10^−4 and 2x10^-4, are not optimal for BERT if it is just trained for 0.5M steps. Because the number of training steps also impact the overall learning rate schedule, I'll need to see results with more peak learning rates. This will help the reader understand the improvement brought by SWE.
> >
> > In addition, two-stage training (first train with sequence length equals to 128 and then switch to 512) can also improve the training speed. Given that there are other ways to reduce the training time and the author just conducted experiments on one hyper-parameter combination: fixed peak learning rate + 0.5M training iterations, I will be hesitant in trying out the method. By reading the paper and the experimental results, I'm not confident about whether SWE can be generalized to other training hyper-parameters (lr + #iter).

---

### Official Review · AnonReviewer3 · 2020-10-28
**Official Blind Review #3**

**Rating:** 6
**Confidence:** 4

**Review:**

This work introduces a method to accelerate the training speed of deep learning models. The key idea is to start training with shared weights and unroll the shared weights in the middle of the training. The authors report that this strategy accelerates the convergence speed. The paper introduces heuristics on when to stop weight sharing and how many layers to share weights. It further provides an analysis via the view of deep linear models on why weight sharing helps improve the convergence speed. In the evaluation, the paper evaluates their approach against the training of BERT, and shows that their method can obtain comparable and sometimes even better accuracy on downstream tasks while with 50% faster training speed.

Strengths:
+ The paper aims to address an important problem in large model training: slow training speed.
+ The paper proposes an easy-to-implement approach to accelerate the convergence speed of BERT-like models.

Weakness:
- The technical contribution seems rather incremental. The main difference between this work and the prior work [1], which also train Transformers via shared weights,  seems to be switching from sharing weights to unsharring weights in the middle of the training.
- Important references are missing, making it not clear the advantage of this work as compared with existing approaches that accelerate the training of Transformer networks.
The comparison with existing work is inadequate. Important ablation studies are needed.

Comments:

Prior work [1] uses weight sharing to train a smaller Transformer model to obtain similar accuracy.  However, weight sharing does not improve training speed per batch, because the training still needs to perform the same amount of forward and backward computation for each batch. Training may actually be slowed down, since the model may need to train with more batches to reach the same accuracy. This paper aims to speed up the training process by switching from shared to unshared weights in the middle of the training, and it observes faster convergence -- achieving similar downstream accuracy with less number of pretraining samples. This is an interesting empirical observation and can potentially become useful in practice. However, there are some major concerns about the paper.

It is still unclear whether this faster convergence comes from switching from sharing to unsharring weights or is an effect of the model or hyperparameter changes. First, the stop condition (e.g., the switching threshold) cannot be known as a prior. Therefore, it is controlled with an additional hyperparameter. From the text, it is unclear how this hyperparameter has been chosen or will be chosen when training a new model. It would be better to test the sensitivity of the hyperparameter on another model such as GPT-2 to verify the effectiveness of the proposed method.

Second, the paper adopts Pre-LN in its evaluation (as briefly mentioned in Section 5.1). However, from the text description, it seems it employs the original BERT as the baseline ("We first show experiment results English Wikipedia and BookCorpus for pretraining as in the original BERT paper"). As Pre-LN has been studied in several prior work [2,3] and has been demonstrated to also have the effect of accelerating the convergence speed, it is unclear whether the observed speedup in this paper is an effect of PreLN or weight sharing/unsharing. An ablation study with BERT-PreLN is needed if not already included.

Third, the analysis on deep linear models appears to be over-simplified, where important characteristics of the DNNs such as non-linear activations and residual branches are not represented, making it difficult to connect it with the actual observations in practice.

Finally, importance reference [4] is missing. [4] starts with a shallow BERT and progressively stacks Transformer blocks to accelerate training speed, which is in some sense similar to the proposed technique, which starts with shallow BERT with shared weights and switches to full-length BERT in the middle of the training. The paper might need to highlight the difference between this work and [4].

Several places in the paper are vague or inconsistent:

1. The paper claims that it uses the same BERT implementation and training procedure as the one used by Devlin et. al.. However, the accuracy reported in Table 2 seems to be consistently lower than what was reported in the original BERT paper. For example, QQP in the original paper reaches 72.1, whereas this paper reports 71.4, QNLI was 92.7 in the BERT paper and 91.7 in this paper. If we use the original BERT reported results, the proposed technique seems to incur low accuracy on most tested datasets. Some clarification on the accuracy results is needed.

2. The paper claims that "it sounds natural for us to expect that ALBERT's performance will be improved if we stop its weight sharing at some point of training. The optimal models are supposed to not be far from weight sharing." However, this explanation actually creates some confusion. First, the paper does not provide an analysis of how the weight distribution between the weights trained with and without weight sharing, so it is unclear what "not be far" means. Second, what does it mean by "optimal model"?  Does it refer to models trained not through weight sharing? If so, prior work [5] identified that model weights and gradients at different layers can exhibit very different characteristics, which seems to contradict the argument that "The optimal models are supposed to not be far from weight sharing".

3. I find it challenging to claim the proposed technique generic for models with repeatedly layers, whereas only BERT is evaluated in the experiments.

4. The paper says "This means that the weight sharing stage should not be too long", but it is unclear how long is considered as not too long.

[1] Lan et. al. "ALBERT: A Lite BERT for Self-supervised Learning of Language Representations", https://arxiv.org/abs/1909.11942

[2] Xiong et. al. "On Layer Normalization in the Transformer Architecture", https://arxiv.org/abs/2002.04745

[3] Shoeybi et. al. "Megatron-LM: Training Multi-Billion Parameter Language Models Using Model Parallelism", https://arxiv.org/abs/1909.08053

[4] Gong et. al. "Efficient Training of BERT by Progressively Stacking", http://proceedings.mlr.press/v97/gong19a/gong19a.pdf

[5] You et. al. "Large Batch Optimization for Deep Learning: Training BERT in 76 minutes", https://arxiv.org/abs/1904.00962

---

> ### Author Response · Authors · 2020-11-23
> **Response to Reviewer #3**
>
> We thank review #3 for the thorough and insightful review. For the concerns raised in the review:
> - On the stop weight sharing condition
>
> For the reported results, we fix $\rho = 0.5$ without tuning for all experiments and it offers good results. In the ablation study, table 3 shows that the performance is not sensitive to the setting of $\rho$ in a large range.
>
> - On the theoretical analysis
>
> We have changed the title of section 3 to theoretical motivation. Our goal here is to provide a motivation for training with weight sharing and using the gradient correlation as the untying criterion. We don’t attempt to provide convergence analysis for BERT-like models.
>
> - On related work
>
> Unlike progressive stacking [4], our approach does not involve any operation that changes the network structure. It achieves acceleration by using a few lines of code for modifying the gradients.
>
> Moreover, progressive stacking [4] requires to predefine when to stack to achieve acceleration. The paper doesn’t provide any guidance on this. Instead, our proposed algorithm can automatically decide when to unshare the weights.
>
> - On the experiment results
>
>   1. Concerns on Table 2 being worse than the original BERT. Results in Table 2 are obtained using the XLNet dataset which is different from the dataset used in the original BERT paper. So they are not comparable. Please see Table 1 for the comparison with the original BERT training.
>   2. Comments on Pre-LN. We updated the submission in which Pre-LN is used in all experiments on all datasets.

---

> > ### Comment · AnonReviewer3 · 2020-11-23
> > **Comments**
> >
> > Thank you very much for addressing my questions! Can you provide more details on how you compare with PreLN than just saying "updated the submission with PreLN in all experiments"? For example:
> > - From the updated results, it seems by using Pre-LN, the gap between the baseline and the proposed technique already becomes smaller. Xiaong et al. (https://arxiv.org/pdf/2002.04745.pdf) find that using PreLN allows one to train BERT with larger learning rates, accelerating the training convergence speed. It appears that you also choose different learning rates for different datasets (3e-4 for wiki + bookcorpus, but 2e-4 for XLNet datasets), but it is unclear why you choose these learning rates and whether you have done some experiments to see if larger learning rates helps further improve the baseline results. Is it possible that the PreLN baseline results are sub-optimal as a result of the smaller learning rates used in the experiments? To be more convincing, it would be better to at least conduct some tuning of the learning rates of Pre-LN to see if the improvement is real or just a use of a weak baseline.
> > - After the revision, it appears that both the baseline and SWE results have improved. For example, in Table 1, the SWE-F, SWE-A, SWE-M results are different from the original version. However, this is a bit confusing, as my understanding was the SWE results already included PreLN and only the baseline results need to include PreLN. Can you explain what changes you did that make the results different?
> > - "Note that we the correct implementation of Pre-LN contains a final layer-norm right before the final classification/masked language modeling layer. " This sentence has a grammatical error.

---

> > > ### Author Response · Authors · 2020-11-23
> > > **Response to Comments**
> > >
> > > Thanks so much for the quick comments.
> > >
> > > --For experiments on wiki+books, we exactly followed the Pre-LN paper by Xiong et al 2020 by setting the peak learning rate to 3e-4 (2x larger than the original BERT's 1e-4). We also tested 2e-4 and 4e-4. Both performed worse than 3e-4.   For the XLNet dataset with whole word masking, however, the peak learning rate of 3e-4 makes training unstable and yields worse performance than 2e-4.
> > >
> > > --In our original submission, on the wiki+books dataset, we did not normalize the input to the last layer (classification/MLM). This normalization is unnecessary for post-LN  but *needed* for pre-LN. In this revision, we fixed this issue and thus all the results are improved.
> > >
> > > --The broken sentence should be “Note that the correct implementation of Pre-LN contains a final layer-norm right before the final classification/masked language modeling layer. “ (That is, removing “we” from the original sentence. )

---

> > > > ### Comment · AnonReviewer3 · 2020-11-23
> > > > **Comments**
> > > >
> > > > Thank you. My concern has largely been addressed, and I increased my score. In practice, one observation is that the unstable training with a larger learning rate can be avoided by tuning the warmup ratio or using optimizers (e.g., RADAM [1]) that reduces the noise in gradients, both may lead to faster convergence speed. The proposed technique accelerates training speed by introducing an additional decision point (switching from weight sharing to unsharing), its impact on the convergence speed is unknown beyond the tested model and datasets. It would be interesting to see in future work the proposed technique to be applied to more variants of Transformer networks.
> > > >
> > > > [1]  Liu et al. "On the Variance of the Adaptive Learning Rate and Beyond", https://arxiv.org/abs/1908.03265

---

> > > > > ### Author Response · Authors · 2020-11-24
> > > > > **Thanks for the reference**
> > > > >
> > > > > We will try the the idea in [1]. It looks very interesting. Next we are gonna conduct experiments on machine translation.

---

### Author Response · Authors · 2020-11-23
**Common Response and Revision**

-- Our goal in this paper is to develop a practically useful fast training algorithm for BERT-like models.  It is *not*  for  theoretical research on BERT or transformer, which is entirely different from and  irrelevant to our focus here. To the best of our knowledge, the presented method should be the simplest solution to speeding up BERT training. All we need to do is to add several lines of code to modify gradients (using gradient mean) to reduce the training time 50%. Please provide feedback around algorithm design & performance,  including clarifying experiment details, missing baselines,  and so on.  That will be most helpful for us to improve this work.

-- The theoretic analysis on deep linear networks is used to *motivate and derive* the adaptive untying criterion used in our training algorithm.   The adaptive untying criterion plays the most fundamental role in the success of our method.  A random chosen untying point has no way to speed up training. The fixed point untying method (Algorithm 1) is used to serve as a baseline for its adaptive version (Algorithm 2). Experiments show that the adaptive method can match (even outperform) the performance of the best manually chosen untying point (ran experiments with different untying points and then chose the best one to compare). This shows our adaptive weight sharing method is effective.

-- Revised the 4th paragraph in the introduction to present a high-level illustration on the connection between the theory and algorithm.  Also Revised the algorithm section to make the connection between the theory and algorithm clearer.

-- Revised the title of the theory section from "theoretic analysis"  to “theoretic motivation”  to avoid potential confusion (e.g. thinks of our paper as an attempt to build theoretic analysis for BERT/transformer). Also revised the first paragraph of the theory section to highlight the goal of our theory.

---

### Decision · Program_Chairs · 2021-01-07
**Final Decision**

**Decision:**

Reject

**Comment:**


The paper proposed a new way for training models that stack the same basic block for multiple times -- share the weights first and then untie the weights. Ablation study shows that the proposed algorithm has marginal improvement over the baseline. The authors also provide some theoretical justifications to how the proposed idea works.
The proposed idea is straightforward and intuitive. Weight sharing has been used in previous works, and what’s new in this paper is to unshare the weights in the middle (with a heuristic rule). The hope is that by doing so, one can achieve a better tradeoff between speedup and accuracy. However the experimental supports are somehow weak and incomplete. For example, in order to show the real speedup, one should provide the full training curve (until convergence) under different settings, instead of just showing one data point (at 500K). It is very common that one can get some speedup at 500K, but the speedup totally disappears after another 500K steps.
Furthermore, the theoretical analysis is conducted in a simplified setting, and it is not very clear whether it can be used to explain what really happened during BERT training.
The reviewers conducted some lengthy discussions after the author rebuttal was available. As a final consensus, we think that there are still concerns on the paper, which makes us hesitate to give an ACCEPT recommendation.